# Drug-Screening Strategies for Inhibition of Virus-Induced Neuronal Cell Death

**DOI:** 10.3390/v13112317

**Published:** 2021-11-20

**Authors:** Durbadal Ojha, Tyson A. Woods, Karin E. Peterson

**Affiliations:** Laboratory of Persistent Viral Diseases, Rocky Mountain Laboratories (RML), National Institute of Allergy and Infectious Diseases (NIAID), Hamilton, MT 59840, USA; durbadal.ojha@nih.gov (D.O.); woodsta@niaid.nih.gov (T.A.W.)

**Keywords:** encephalitic virus, neuronal cell lines, drug screen, cytotoxicity concentration, effective concentration, selectivity index

## Abstract

A number of viruses, including Herpes Simplex Virus (HSV), West Nile Virus (WNV), La Crosse Virus (LACV), Zika virus (ZIKV) and Tick-borne encephalitis virus (TBEV), have the ability to gain access to the central nervous system (CNS) and cause severe neurological disease or death. Although encephalitis cases caused by these viruses are generally rare, there are relatively few treatment options available for patients with viral encephalitis other than palliative care. Many of these viruses directly infect neurons and can cause neuronal death. Thus, there is the need for the identification of useful therapeutic compounds that can inhibit virus replication in neurons or inhibit virus-induced neuronal cell death. In this paper, we describe the methodology to test compounds for their ability to inhibit virus-induced neuronal cell death. These protocols include the isolation and culturing of primary neurons; the culturing of neuroblastoma and neuronal stem cell lines; infection of these cells with viruses; treatment of these cells with selected drugs; measuring virus-induced cell death using MTT or XTT reagents; analysis of virus production from these cells; as well as the basic understanding in mode of action. We further show direct evidence of the effectiveness of these protocols by utilizing them to test the effectiveness of the polyphenol drug, Rottlerin, at inhibiting Zika virus infection and death of neuronal cell lines.

## 1. Introduction

Viral-induced encephalitis is a rare, yet serious, neurological disease that can be life-threatening. A number of viral families can cause viral encephalitis including Herpesviruses (particularly Herpes Simplex Virus 1 (HSV-1) as well as HSV-2, Cytomegalovirus, Epstein-Barr virus, Varicella-Zoster virus), Picornaviruses (Enterovirus D68, Poliovirus), Paramyxoviruses (Measles virus, Nipah virus), Flaviviruses (West Nile virus, Japanese encephalitis virus, Zika virus), Bunyaviruses (La Crosse Virus, Tahyna virus, Jamestown Canyon virus), Togaviruses (Eastern and western equine encephalitis virus, St. Louis encephalitis virus, Rubella virus), Rhabdovirus (Rabies virus) and Reoviruses (Colorado tick fever virus) [1,2,3,4,5,6,7,8,9,10,11,12,13]. Many of these viruses directly infect neurons in the CNS and can induce neuronal damage either by causing neuronal death or neuronal dysfunction [14,15,16,17]. Inhibiting the ability of these viruses to infect or kill neurons may be one of the key factors in being able to therapeutically treat viral encephalitis. Currently, there is a lack of therapeutic treatment for most cases of viral encephalitis [18,19]. Clinical management of most viral encephalitis cases is primarily limited to palliative care [20,21]. One of the few exceptions is the nucleoside analog drug, Acyclovir, which has been used to treat HSV-1, albeit with limited efficacy [22,23]. Thus, there is a strong need to identify and develop drugs that can effectively treat viral encephalitis.

One mechanism to identify potential candidate therapeutics is through drug screens of established drug libraries such as the NCATS Pharmaceutical Collection (NPC) of FDA-approved drug library [24]. Established libraries can be screened with a range of drug concentrations for the effectiveness of each drug against virus-induced cell death using a readout of cell viability or virus production [24,25]. This is generally done using cell lines expressing a fluorescent or luciferase reporter readout or measuring virus-infection using a fluorescent or luciferase-tagged virus [24,26]. Other assays that are effective but do not necessarily have the same high-throughput capability are those that measure cell death such as apoptosis assays, dye-uptake cell viability assays such as MTT (3-(4,5-dimethylthiazol-2-yl)-2,5-diphenyl tetrazolium bromide) or XTT (2,3-bis(2-methoxy-4-nitro-5-sulfophenyl)-5-carboxanilide-2H-tetrazolium) assays or using an instrument which directly counts the number of alive cells such as a hemocytometer or cell counter [27,28,29,30].

In this manuscript, we discuss key considerations for analysis of compounds selected from an initial screen for encephalitic viruses. We discuss and provide detailed methodology of useful cell lines and primary cells as well as determining the efficacy and selective index for each compound and determining the direct mechanism of drug inhibition of the virus. Finally, we provide “proof of principle” data showing the ability of the drug, Rottlerin, to inhibit Zika virus (ZIKV)-induced damage of neurons. ZIKV is an arbovirus belonging to the family of Flaviviridae. Human transmission can occur after the bite of an infected Aedes species mosquito (*Ae. aegypti* and *Ae. albopictus*). However, it causes mainly asymptomatic to mild symptomatic infection with fever, headache, conjunctivitis, muscle and joint pain, etc., in adults. In some cases, it causes neurological diseases such as Guillain-Barré syndrome (GBS) in adults and fetal abnormalities such as microcephaly and other severe brain defects in newborns [8]. To date, there are no licensed vaccines or therapeutic drugs to treat ZIKV-induced neurological disease. In this study, we utilize several of the below protocols to analyze the efficacy of Rottlerin, a polyphenol recently found to inhibit peribunyavirus replication in neurons [24], against ZIKV-induced neuronal damage.

## 2. Primary Neurons and Neuronal Cell Lines for Use in Testing Candidate Therapeutics

One of the important aspects is selecting cells or cell lines which were based on organ specificity of the virus infections and its ability to kill and replicate. There are several useful neuronal cell lines for testing the effect of drug inhibition of virus-induced cell death in neurons, including SH-SY5Y cells that have been well characterized [31], Neuro-2a, a mouse neuroblastoma cell line (ATCC CCL-131), and C17.2 cells, a retrovirus-transformed murine neural cell line [32]. Subsequent analysis of candidate compounds can then be completed on additional cell lines as well as human stem cells or mouse primary cortical neurons (described below) [33] to determine if the target drug consistently inhibits virus infection or virus-induced cell death, before testing in advanced model systems. Below is detailed information on several cell lines that can be used to study neurovirulent viruses. It is important to test these cell lines for ability to be infected with the virus being studied and to do a kinetic analysis of virus replication in the key cell lines.

### 2.1. Primary Cortical Neuron Isolation

#### 2.1.1. Instruments/Materials for Isolation of Neurons

Magnifying glass with light source/dissecting scopeSmall scissorsTwo pairs of sharp #5 forcepsAbsorbent padsAbsorbent wipes15 and 50 mL polypropylene centrifuge tubes10 mL pipets1 mL syringes20 g needlesSterile transfer pipettesSterile 100 mL beaker for keeping sterile instrumentsIce buckets and trayBiohazard bags10× PBS pH 7.412% D-glucose (20×) in PBS (filter sterile)6-, 12- and 24-well amine-coated tissue culture plates0.4% Trypan blue solution

#### 2.1.2. Essential Media and Reagents


*Neurobasal medium 1×, 500 mL*
Aliquot in 50 mL and store at 4 °C in darkNeurobasal media with B27 supplement and 0.5 mM L-glutamine

*B-27 serum-free supplement 50*
*×, 10 mL*

*Glutamax I (200 mM, in 0.85% NaCl) 100 mL*

*Calcium, Magnesium Free (CMF)-Hanks Balanced Salt Solution (HBSS) 10× liquid 500 mL*
KCl (4000 mg/L), KH_2_PO_4_ (600 mg/L), NaCl (80,000 mg/L), Na_2_HPO_4_.7H_2_O (900 mg/L), D-glucose (10,000 mg/L)HBSS

*HEPES buffer solution 100*
*× (1 M)*
238.3 g/L of HEPES prepared in distilled water. pH: 7.2–7.5

*Trypsin, 2.5% (10×) 100 mL*
25 g of trypsin and 8.5 g of NaCl in 100 mL


#### 2.1.3. Solutions or Media to Prepare Just before Use

*Dissection medium* (100 mL): CMF-HBSS (Ca^2+^, Mg^2+^ free Hank’s Balanced Salt Solution) buffered with 10 mM HEPES, pH 7.310 mL 10*×* CMF-HBSS1 mL 1 M HEPES (100×)89 mL H_2_ODigestion solution (5 mL)4.5 mL CMF-HBSS with HEPES0.5 mL 2.5% Trypsin
Add 4.5 mL of CMF-HBSS with HEPES in 15 mL tubes and keep at room temp. Do not add trypsin before use.Keep 2.5% trypsin tube at 37 °C water bath.*Neuronal plating medium* (100 mL)90 mL Opti-MEM with L-glutamine5 mL D-glucose 12% (20×) filter sterilized5 mL FBS (5% final conc.)*Neuronal maintenance medium* (100 mL)98 mLNeurobasal medium2 mL B-27 supplement 50*×* (2% final conc.)250 µL Glutamine 200 mM (0.5 mM final conc.)

#### 2.1.4. Bench Preparation

Clean the workbench, dissecting microscope and mouse pad thoroughly with 70% ethanol.Place instruments in 70% ethanol for several minutes to sterilize. Lightly shake off ethanol from instruments and move them to an empty sterile 100 mL beaker. Let them air dry completely before use.Fill ice bucket and kidney tray (for keeping petri dishes ice cold) with ice.Prepare 15 mL polypropylene tubes with 4.5 mL of dissection medium and let them sit at room temp.Add 30–35 mL of ice-cold sterile dissection medium in petri dishes (100 mm) and let them sit on ice in kidney tray (at least 4 dishes).Warm 2.5% trypsin to 37 °C and neuron plating media to room temperature.

#### 2.1.5. Protocol for Isolation and Culturing

Euthanize a pregnant mouse (14–17-day gestation) by Isoflurane anesthesia and exsanguination (as defined by an accepted Animal Care and Use protocol). Dissect out the entire uterus along with embryos and place in a sterile petri dish containing ice-cold dissection medium. Keep petri dish on ice in a kidney tray.Remove the fetuses from the uterus, and place in a fresh petri dish with ice-cold dissection medium. Separate the head from the body by pulling. The tissue must always be kept submerged in the dissection medium.Remove the skin from the head and open the skull cap with forceps. Remove the brain and separate from the rest of the tissue. Separate the cortices from the brain and remove the meninges from the cerebral cortices.Collect all the cortices in a 15 mL conical centrifuge tube containing 4.5 mL dissection medium (at room temp).Add 0.5 mL of 2.5% trypsin and incubate for 15 min in a water bath at 37 °C (invert tube halfway through).Gently remove supernatant with a sterile transfer pipet, leaving cortices at the bottom of the tube.Bring the volume to 15 mL with fresh dissection medium, invert the tube 3–4 times gently and let it stand for 5 min at room temperature. Repeat this to allow residual trypsin to diffuse from the tissue. Bring the final volume to 4 mL with neuron plating medium.Dissociate the cortices by gently pipetting up and down 10 times with a 10 mL pipet. Split into two 15 mL tubes, each with 2 mL of rough homogenate in them. Dissociate rough homogenate by putting through a 27-gauge needle on a 1 mL syringe approximately 10 times.Bring the volume to 10 mL with neuron plating medium.Count the total number of cells: Add 10 µL of cell suspension and 10 µL of 0.4% filtered trypan blue in a 96-well plate, mix them well and load on a Neubauer chamber slide and count number of live cells in each corner square and take the average number per corner square and calculate total number of cells per mL using the formula.
Cells/mL = Average cells counted in corner square × dilution factor × volume conversion factor (10^4^)Plate cells on amine-coated plate using neuronal plating medium.After 3–4 h, examine the dishes to ensure that most of the cells are attached, then aspirate the entire medium from all the wells and replace with neuronal maintenance medium.Thereafter, replace 1/2 of the media with fresh neuronal maintenance media once per week.

### 2.2. Culture of Neuronal Cell Lines

#### 2.2.1. Mouse Neuroblastoma Cell Lines

*i* 
*Neuro-2a*
*ii* 
*C17.2*


#### Essential Media and Reagents

Dulbecco’s Modified Eagle Medium (D-MEM) containing 4500 mg/L D-glucose, L-glutamine and 110 mg/L sodium pyruvate including 10% fetal bovine serum (FBS) and 1× Penicillin−Streptomycin solution.0.25% Trypsin with EDTA.

#### 2.2.2. Human Neuroblastoma Cell Lines


*SH-SY5Y*


#### Essential Media and Reagents

Eagle’s Minimum Essential Medium (EMEM) and F-12K (Kaighn’s Modification of Ham’s F-12) medium in equal amounts (1:1) supplemented with 10% FBS and 1× Penicillin−Streptomycin solution.0.25% Trypsin with EDTA.

#### Thawing of Cell Lines

These cells are quite fragile. Do not centrifuge immediately after thawing. Gently pipette contents of vial into 6-well plate. Plate two-fold dilutions of the cells: 500, 250, 125 µL with 3 mL of media per well. The media can then be replaced in 4–8 h after the cells adhere to remove the freezing media.

#### Sub-Culturing

These cells grow as a mixture of floating and adherent cells. Remove the medium with the floating cells.Rinse the adherent cells with fresh 0.25% trypsin 0.53 mM EDTA solution, add an additional 1 to 2 mL of trypsin solution and let the culture sit at room temperature (or at 37 °C) until the cells detach, and recover the cells by centrifugation (5 min at 500 Gs).Add fresh medium, aspirate, combine with the floating cells recovered above and dispense into new flasks. An inoculum of 3 × 10^3^ to 1 × 10^5^ cells/cm^2^ is recommended.Subculture when cell concentration is between 8 × 10^5^ to 1 × 10^6^ cells/cm^2^ (~80% confluency).

#### Culture Conditions

Atmosphere: air, 95%; carbon dioxide (CO_2_), 5–7%; Temperature: 37 °C.

#### 2.2.3. Human Neuronal Stem Cell Line (NSC)

##### Essential Media and Regents

StemPro^®^ NSC SFM complete medium (100 mL):KnockOut™ D-MEM/F-12 97 mLStemPro^®^ NSC SEM, 200 mM 2 mLGlutaMAX ™-I Supplement 1 mLRecombinant Human Fibroblast Growth Factor Basic 20 µLEpidermal Growth Factor 20 µLFibronectinPhosphate-Buffered Saline without Ca^2+^ and Mg^2+^ (DPBS)TrypLE™ Express Stable Trypsin Replacement Enzyme

##### Coating Culture Vessels with Fibronectin

Dilute fibronectin in distilled water. Prepare 1 mg/mL stock solution. Store at −20 °C.Dilute stock solution in D-PBS to make working solution of 20 µg/mL.Add sufficient amount of working solution to the culture flask or plate to cover the bottom surface of the culture vessels (8–10 mL for T-75 flask or 1 mL per well of 6-well plate or 0.5 mL per well of 12- and 24-well plate.)Incubate for 1 h at 37 °C in humidified atmosphere of 5% CO_2_.Remove the vessel and store it at 4 °C until use.Discard Fibronectin solution immediately before use and wash with PBS.

##### Thawing of Cell Line

Warm 10 mL of KnockOut™ D-MEM/F-12 media in 15 mL centrifuge tube at 37 °C.Remove the frozen vial of NSC from nitrogen tank. Immediately transfer to water bath.Mix the thawed cells into prewarmed media in a 15 mL centrifuge tube. Spin down the cells at 500 G for 4–5 min.Resuspend cells in StemPro^®^ NSC SFM complete medium at 10^5^ cells/mL and culture in a T-25 or T-75 flask coated with fibronectin as described in the previous section.

##### Culture Procedure

When cells are about 80% confluent, aspirate the medium and wash with DPBS.Add 1 mL of TrypLE™ Express to the flask and incubate at room temperature (or 37 °C) until the cells detach. Add 5–6 mL of fresh media and recover it in a 15 mL centrifuge tube.Spin down cells by centrifugation at 500 G for 4–5 min.Count cell numbers using hemocytometer as described in Section 2.1.5.Plate cells in fresh StemPro^®^ NSC SFM complete medium at specific cell count for experiments on fibronectin-coated plates.

## 3. Virus Infection Procedure

Warm virus at room temperature or 37 °C for thawing. When thawed, place on ice.Dilute the virus in neuronal maintenance media based on final volumes below (iii). A multiplicity of infection (MOI) of 0.01–10.0 often works well for virus quantification and MTT/XTT assays. Keep dilution on ice.Add virus containing media to wells using below final volumes:
For 6-well plate: 3 mL per well.For 12-well plate: 1–2 mL per well.For 24-well plate: 0.5–1 mL per well.For 48-well plate: 200–400 µL per well.For 96-well plate: 100–200 µL per well.Incubate at 37 °C for 1 h for virus attachment.Remove virus-containing media, wash with PBS and add fresh media at same volumes as described above.Place plates in 37 °C CO_2_ incubator for 2–5 days until 80–100% CPE.

### Selection of Cell Types for Drug Study against Zika Virus (ZIKV)

Multiple cell lines were analyzed for ZIKV infection and cell death (Figure 1A). Cells were plated in 96-well plates using 1 × 10^4^ to 4 × 10^4^ cell/well depending upon cell size and infected at an MOI of 1.0 following the above protocol. At 4 dpi, cells were examined by light microscopy for cytopathic effect (CPE) as well as for cell viability using MTT assay (Section 4, below). The initial cell types screening results showed that SH-SY5Y and hNSCs produced CPE upon ZIKV infection (Figure 1A). ZIKV induced approximately 80−90% cell death in hNSCs and 50–60% cell death in SH-SY5Y. No ZIKV-induced CPE or cell death was observed with mouse primary cortical neuron, N2a or C17.2 cells. Thus, we chose hNSCs for therapeutic drug studies. Comparison of 0.1, 1 and 10 MOIs for CPE and cell viability showed substantial cell death with 1 or 10 MOI but not 0.1 MOI (Figure 1B,C). Further studies were done with 1 MOI.

## 4. Screens of Selected Compounds to Determine Efficacy

One of the important considerations in analyzing therapeutic compounds is their Selectivity Index (SI), the comparative effectiveness of the specific drug for inhibiting cell death relative to the toxicity of the compound for the same cell. As many drugs can induce host cell apoptosis or cell death alone, it is necessary to analyze increasing concentrations (two-fold dilution) of the drug in question to determine the median cell cytotoxicity concentration (CC_50_), the concentration that results in 50% cell death of the host cell [24,34]. The effective concentration (EC_50_) should also be measured by serial dilution as the concentration of the drug that produces 50% inhibition of virus-induced cell death [24,35,36]. The SI is then calculated by dividing the EC_50_ by the CC_50_ [24,37,38]. An SI that is substantially higher, i.e., a score of 10 or more, would be indicative of a drug that may provide therapeutic benefit and may be worthwhile to analyze in more advanced model systems such as organoid cultures or in vivo models of disease. The USA Food and Drug Administration (FDA) requires the evaluation of cytotoxicity and anti-viral efficacy of potential candidate drugs before any clinical testing [38].

### 4.1. In Vitro Anti-Viral MTT or XTT Assay

MTT or XTT assays can be used as a sensitive assay to measure cell death. In this assay, cell viability is measured by the formation of purple insoluble formazan crystals from yellow water-soluble dye MTT or XTT inside the viable cells by mitochondrial enzymes. Cytotoxicity of the candidate compounds as well as virus-induced cell death can be evaluated using MTT/XTT assays (Figure 2). Thus, drug efficacy versus drug-induced cytotoxicity can be measured with the same assay. For example, the median EC_50_ of Acyclovir was 0.91 μg/mL by calculation inhibition of 50% virus-induced cytopathicity when it was tested against 351 isolates of HSV-2 [39]. Of them, seven isolates had EC_50_ values ≥ 3 μg/mL, which were considered Acyclovir resistant. The drug Chloroquine, a 4-aminoquinoline and an FDA-approved drug to treat malaria, was shown to inhibit ZIKV in vitro [40,41]. The EC_50_ concentration of chloroquine against ZIKV infection was 9.82–14.2 μM as assessed by cell viability using Vero cells, human brain microvascular endothelial cells (hBMECs) and NSC [41].

#### 4.1.1. MTT or XTT Reagent Preparation

Stock concentration: 5 mg/mL in PBS. Dissolve 50 mg of MTT or XTT powder in 10 mL of PBS. Keep on magnetic stirrer with stir bar for 5–10 min to dissolve these reagents completely.Sterile filter through 0.22 µm filter. Protect the reconstituted MTT or XTT reagent from light and store at 4 °C.STOP solution: Use DMSO or 20% SDS in 50% dimethylformamide. Store at room temperature in fume hood.

#### 4.1.2. Assay Protocol for Cytotoxicity

Culture cells in 48- or 96-well plate using 10^3^ to 10^5^ cells per well (~80% confluent) and incubate for 4–6 h in 37 °C CO_2_ incubator.Remove the media and add fresh media containing two-fold dilution of the test drug.After 2 to 4 days incubation, remove media and wash with PBS (1×).Add fresh media containing 1/10 total volume of MTT stock reagent to each well.Incubate the plate at 37 °C in dark for 3–4 h. (Reduce incubation times to 2 h if cell density is high.)Add 100 to 200 µL of STOP solution into each well and place on shaker for 30 min to 1 h at room temperature to lyse the cells and dissolve the formazan completely. (Alternatively, you can pipet up and down several times to dissolve formazan but avoid air bubbles).Transfer 100 µL of the homogenous solution from each well into 96-well plates and measure the absorbance on an ELISA plate reader at 540 nm.Determine the percent viability of cells = (mean absorbance of drug treated sample/mean absorbance of control) × 100.Determine CC_50_ (the cytotoxic concentration of the drugs to cause death of 50% of viable cells) by extrapolating dose−response curve.
After determining the CC_50_, this concentration should be used as highest dose of the drug for efficacy study. Starting at this initial highest dose, use two to ten (log)-fold dilutions of decreasing concentrations of the drug for dose–response curve in efficacy study.

#### 4.1.3. Assay Protocol for In Vitro Efficacy

Culture cells in 48- or 96-well plate using 10^3^ to 10^5^ cells per well (~80% confluent) and incubate for 4–6 h in 37 °C CO_2_ incubator.Remove the media and infect with virus at 0.01–10.0 MOI. Add two to ten-fold dilutions of the test drug to wells and incubate 1 h.Remove the media and wash with PBS.Add fresh media (containing the same amount of test drug as before) to each well and incubate 2–7 days.After 2–7 days of incubation, perform MTT/XTT assay (assay protocol for cytotoxicity) starting at step iii.After calculating the percentage of viable cells, determine the EC_50_ or IC_50_ and SI (CC_50_/EC_50_) from the dose–response curve.


**Anti-ZIKV activity of Rottlerin (RTL)**


RTL is a polyphenol, first isolated from Asian plant *Mallotus philippinensis*. A previous report showed that RLT has a variety of beneficial potency such as an antioxidant [42], anti-inflammatory [43,44], antiallergic [44], antibacterial [45,46] and anticancer activity [47,48]. RTL is a very versatile substance that has been used as a selective inhibitor of PKC-δ [49,50]. We recently showed that RTL inhibits La Crosse Virus (LACV)-induced cell death and virus replication in both human and murine neuronal cell lines [24]. To examine if RTL could also inhibit ZIKV, we tested the efficacy against ZIKV infection on hNSCs and Vero cells the above MTT assay. Cytotoxicity concentration of 50% (CC_50_) for RTL was determined by plotting a dose–response cell survival curve (Figure 3A,C). CC_50_ of RTL was 2.37 ± 0.14 and 4.1 ± 0.32 µg/mL for hNSCs and Vero cells, respectively. To determine 50% effective concentration (EC_50_), a similar assay was performed in the presence of ZIKV (Figure 3B,D). The EC50 of RLT were 0.14 ± 0.03 and 0.48 ± 0.08 µg/mL in respective cell lines. The calculated selective index (SI) was 16.9 for hNSCs and 8.5 for Vero cells. Thus, RTL has a high SI to inhibit ZIKV-induced cell death in multiple cell lines.

### 4.2. Determination of Intra-Cellular or Extra-Cellular Virus by Plaque Assay from Drug-Treated Cells

Viral plaque assays can be used to determine the virus titer within cells or in cell supernatant in infected or drug-treated cultures (Figure 4) [24,51]. In this assay, an infectious virus replicates to form a localized zone of infected or dead cells and form a ‘plaque’ which is detected by specific cellular stains. This method also helps to calculate the Inhibitory (effective) concentration 50% I(E)C_50_ measured by serial dilution as the concentration of the drug that produces 50% inhibition in virus production or replication. The I(E)C50 can vary substantially between cell lines. This was observed with the nucleoside analog sofosbuvir, a commercially available RdRp inhibitor, which has been used to treat chronic HCV infection [52]. Sofosbuvir inhibited ZIKV replication with I(E)C50 range of 0.4–5 µM in SH-SY5Y, human placental choriocarcinoma (Jar) cells, and human hepatocellular carcinoma (Huh-7) cells. However, a much higher concentration for I(E)C50 of 32 µM was sound when tested in human fetal-derived hindbrain and cerebral cortex neuronal stem cells (NSCs) [53,54,55].

Assay protocol for intra-cellular or extra-cellular virus count:

Plate cells in 12- or 24well plate using 10^4^ to 10^6^ cells per well depending on well size (60–80% confluent). Incubate for 4–6 h in 37 °C CO_2_ incubator.Remove media and add fresh media containing virus at an MOI of 0.01–10.0 MOI. Add desired concentration of the test drug to wells and incubate 1 h.Remove media and wash with PBS or fresh media.Add fresh media containing same concentration of drug and incubate until desired time points.

#### 4.2.1. For Extra-Cellular Virus

At specific time intervals, remove 100 µL of aliquot of cell supernatant, centrifuge to remove any cell debris and store supernatant at −80 °C. Add 100 µL of fresh media (containing test drug) and incubate until next time point. Analyze for virus using plaque assay described below.

#### 4.2.2. For Intra-Cellular Virus

At desired time interval, remove the media and wash wells with fresh media.Remove cells by trypsin or scrapping and centrifuge.Freeze the cell suspension in dry ice or −80 °C freezer to disrupt the cell membrane.Prior to plaque assay, thaw the cells using a 37 °C water bath and vortex.

#### 4.2.3. Reagent Preparation for Plaque Assay

10% FBS media: DMEM 500 mLPen/Strep 5 mLFBS 50 mL2% FBS DMEM: DMEM 500 mLPen/Strep 5 mLFBS 10 mL1.5% CMC: In a sterile hood, add 7.5 g of sterilized CMC sodium salt (see below) to 500 mL bottle of MEM media.Shake at 37 °C until CMC is dissolved. Sterile CMC sodium salt (Sigma Aldrich C4888): Measure 7.5 g into 20 mL beaker and cover with aluminum foil.Sterilize in autoclave at 250 °F for 20 min and dry cycle for 30 min.10% formaldehyde: 32% formaldehyde 135 mL.Water 365 mL.0.35% crystal violet: crystal violet 1.7 g into 95% ethanol (75 mL) until dissolved.Bring volume to 500 mL with nanopure water. Can be stored at room temperature indefinitely.

#### 4.2.4. Procedure

Grow Vero cells to confluency in T-25; trypsinize and resuspend cells in total 12.5 mL of 10% FBS media. Add 0.5 mL to each well of 24-well plate and incubate at 37 °C incubator. Can fill just one well and check density before doing entire plate. Keep cells well mixed in flask while dispensing.Allow plates to become confluent.Viral dilutions: Add 900 µL of 2% FBS-DMEM to small snap-cap tubes. Add 100 µL of sample to first tube, mix well, change pipette tip and transfer 100 µL to second tube. Repeat this process until desired dilutions are reached. Dilutions of 10^−1^ to 10^−8^ are usually sufficient to determine the amount of virus in each sample. Always run a positive control (known concentration of virus) and a negative control (media alone) for each assay.NOTE: If you want to conserve virus, make first dilution in 450 µL of media using 50 µL of virus. Rest of dilution scheme is the same. This dilution is rarely plated.Discard media from plated Vero cells into an appropriate waste container, leaving small amount of media to keep cell monolayer from drying out.Starting from highest dilution, add 200 µL of diluted sample to each of three wells. Mix tube well before adding media/virus to the cells.Place in incubator for 1 h for adsorption and infection.Add 0.5 mL of warm 1.5% CMC directly to wells with media/virus. Place back in incubator for 5 days.NOTE: Do not touch the pipette to the wells while adding CMC. Add CMC from a height, otherwise you will contaminate the CMC in the bottle with virus.After 5 days, add 10% formaldehyde directly to fill wells, incubate at room temperature for at least 1 h.Pour off formaldehyde solution into appropriate container, rinse wells in tap water several times to remove all CMC. Do not spray directly into wells.Add enough 0.35% crystal violet to cover bottom of each well, and let sit for 10 min.Pour off stain, and again wash gently with tap water as in step ix.Allow to dry at room temperature.Plaque counting:
○Pick dilutions where plaques can be easily distinguished from each other for counting.○Calculations: Virus titer (PFU/mL) = Number of Plaques x dilution factor x infection volume factor.○Infection Volume factor:For, 200 µL = 5For, 100 µL = 10


**Extra-cellular and intra-cellular virus titer determination upon RTL treatment**


We examined whether RTL inhibited ZIKV replication in hNSCs (Figure 5). Virus titers were quantified to analyze the (Figure 5A) release of infectious virus in the cell supernatant (extracellular) and (Figure 5B) mature virus particles inside the cells (intra-cellular) using hNSCs. The plaque assay revealed that an EC_50_ or 2xEC_50_ dose of RTL significantly reduced virus production in cell supernatant (extra-cellular) by up to 2 logs (Figure 5A) and intra-cellular infectious viruses by up to 1 log (Figure 5B).

## 5. Basics in Mode of Action Study

### 5.1. Determining When the Drug Affects Virus Infection: Time of Addition Assay

Time of addition assay is used to investigate the optimal time of drug addition either prior to infection, during infection or during replication [24,56]. This assay helps determine the basic mode of action of a drug on the stages of virus infection, replication and production (Table 1).

#### Assay Protocol for Time of Addition Assay


*Pre-treatment:*
Culture cells in 48- or 96-well plate at 10^4^ to 10^5^ cells per well and incubate for 6–9 h in incubator containing 5% CO_2_ at 37 °C.Remove the media and add fresh media containing 2- to k10-fold higher EC_50_ concentration of the test drug.After 1–3 h incubation, remove the media containing test drug and wash with PBS.Infect with virus at 0.01–10.0 MOI and incubate for 2–7 days.Read out of MTT/XTT assay and/or virus titers.

*Co-treatment:*
Culture cells in 48- or 96-well plate using 2 × 10^4^ or 1 × 10^4^ cells per well and incubate for 6–9 h in incubator containing 5% CO_2_ at 37 °C.Remove the media and infect with virus at 0.01–10.0 MOI as well as add 2-to-10-fold higher EC_50_ concentration of the test drug and incubate for 1 h.Remove the media containing unbound virus and wash with PBS.Add fresh media (without drug) and incubate for 2–7 days.Read out of MTT/XTT assay and/or virus titers.

*Post-treatment:*
Culture cells in 48- or 96- well plate using 10^3^ to 10^6^ cells per well and incubate for 6–9 h in incubator containing 5% CO_2_ at 37 °C.Remove the media and infect with virus at 0.01–10.0 MOI.Remove the media containing unbound virus and wash with PBS.Add fresh media containing 2-to-10-fold EC_50_ of test drug and incubate for 2–7 days.Read out of MTT/XTT assay and/or virus titers.
*Throughout-treatment*: This is the combination of previously described three treatment regimens (Pre-, Co- and Post-treatment with drug).

#### 5.2. Virus Attachment Assay

The aim of this assay is to examine if the drug inhibits virus entry to the cells by neutralizing either virus or host cell receptor [29,56].

##### Assay Protocol for Virus Attachment Assay

Grow cells in 12- or 24- well plate depending on well size.Incubate overnight.Prechill cells at 4 °C for 1 h.Mix same volume of the drug (at 4× EC_50_ concentration) and virus (2× specific MOI).Add drug and virus mixture to the cells and incubate for 3 h at 4 °C or at top of ice.Wash two to three time with ice-cold PBS to remove unbound virus.Add fresh medium to the cells and incubate for either an early time point (1 to 2 h) or late time point (24 or 48 h).For early time point: Remove supernatant and perform qRT-PCR or immunohistochemistry to detect efficacy of the drug.For late time point: Harvest cell supernatant after desired time points and titrate the viruses using previously described plaque assay.
Use heparin sodium salt as a positive control [24,56].

### 5.3. Viral Inactivation Assay

The purpose is to determine if the drug directly affects the virus prior to cell infection (Figure 6) [29,56].

#### 5.3.1. Assay Protocol for Viral Inactivation Assay

Mix the drug (at 2–10-fold EC_50_ concentration) with virus (10^3 to^ 10^6^ PFU/mL) and incubate for 1 h in 37 °C.Dilute 100-fold (10–1000 PFU/well) with media containing 2% FBS to get subtherapeutic concentration of the drug.

#### 5.3.2. Direct Method

Titrate the viruses using previously described plaque assay.

#### 5.3.3. Indirect Method

Plate cells in 12- or 24-well plate using 10^4^ × 10^6^ cells per well depending on well size.Incubate cells overnight.Remove media and add diluted virus and drug mix and incubate 1 h.Remove media and wash with PBS or fresh media.Add fresh media and reincubate cells in incubator.Harvest cell supernatant after desired time pointsTitrate the viruses using previously described plaque assay.
The 100-fold dilution helps to minimize the effect of the drugs in its effective dose and prevent significant interaction with host cells.Use neutralizing antibody as a positive control [24].


**Mode of action studies of RTL**


Since RTL inhibited ZIKV production and cell death, we performed a time of addition assay to understand the basic antiviral mechanism and the timeframe of RTL’s effect on ZIKV infection. hNSCs cells were treated with RTL (2xEC50) at pre-treatment, co-treatment, post-treatment or throughout-treatment and quantified the cell viability using MTT assay. RTL pre- and co-treatment did not protect against ZIKV (Figure 7). However, adding RLT at post and throughout time frames did significantly increase cell viability (Figure 7). This study suggests that the RTL inhibits ZIKV virus infection after entry to the cells, possibly by preventing virus replication or release. Further studies, including analysis of cellular localization of virus in RTL-treated cells, would be needed to directly determine the mode of action of inhibition. However, this study indicates that RTL prevents ZIKV-induced cell death by limiting virus replication in neuronal stem cells. Furthermore, this inhibition occurs post infection of the cell.

## 6. Conclusions

The development of therapeutics that inhibit virus replication has often been difficult with only a limited number of compounds that have shown efficacy in limiting viral pathogenesis. Gaining a better understanding of the mechanism of drug inhibition as well as the timeline of when and how a compound inhibits virus replication in cell lines may help in improving the identification of key therapeutics. This is particularly important for diseases such as viral encephalitis or microcephaly, where a useful therapeutic would be required to inhibit virus infection and/or damage in the cells infected within the CNS, such as neurons. As cell types differ in their metabolism, analysis of the cell type of importance is critical to determine potential effectiveness of a compound. Of course, once clear candidate compounds have been identified by these studies, further studies with more complex systems including organoids and animal models need to be completed. However, the above strategy should help define which compounds should be followed up with those types of studies. Indeed, the current studies show that RTL, a polyphenol drug previously shown to inhibit La Crosse virus [24], also inhibits ZIKV-induced death of neurons. Possibly, RTL may be a useful compound to inhibit other encephalitic viruses as well.

Overall, this paper presents a comprehensive outline for testing small molecules or drugs for their efficacy over cell viability and viral replication. The methodology presented here includes detailed information on cell-based screening, validation and identification of basic mode of action of small molecules/drugs that can be easily extended into several other viruses by selecting relevant and susceptible cell lines [36,57,58,59].

## Figures and Tables

**Figure 1 viruses-13-02317-f001:**
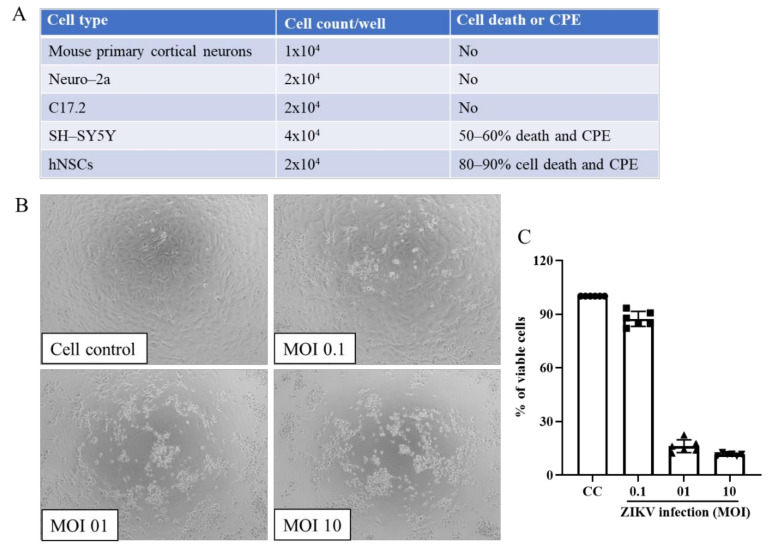
Selection of appropriate cell lines and virus MOI for ZIKV infection. (**A**) Screening of multiple neuronal cell lines against ZIKV infection. (**B**) Light microscopic images of MOI-based ZIKV induced cytopathic effect (CPE) on hNSCs. (**C**) MOI-dependent cytotoxicity of ZIKV infection by MTT assay. Each bar represents the mean ± SD of two independent experiments with three replicates per experiment.

**Figure 2 viruses-13-02317-f002:**
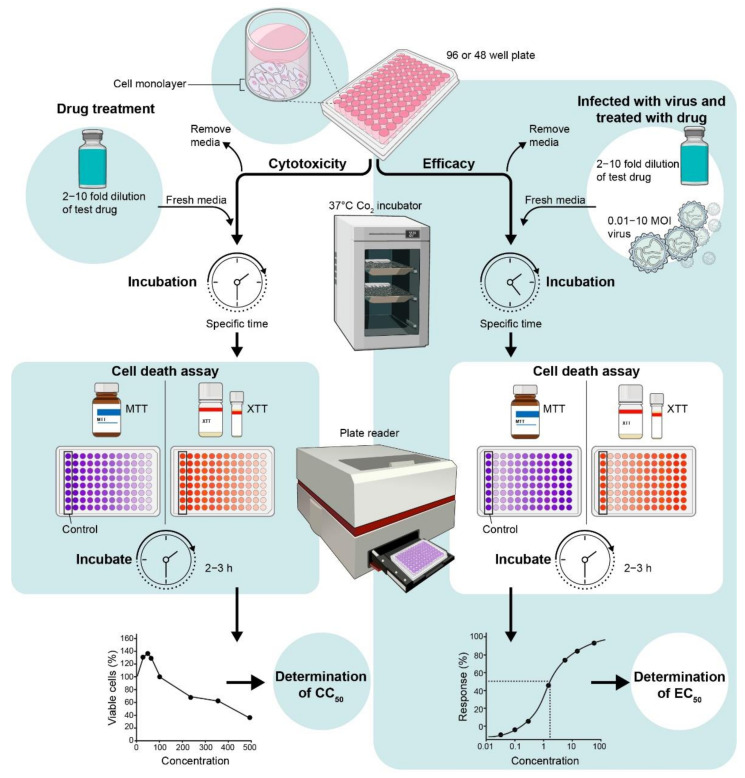
Schematic diagram of screening for drug activity against virus-induced cell death.

**Figure 3 viruses-13-02317-f003:**
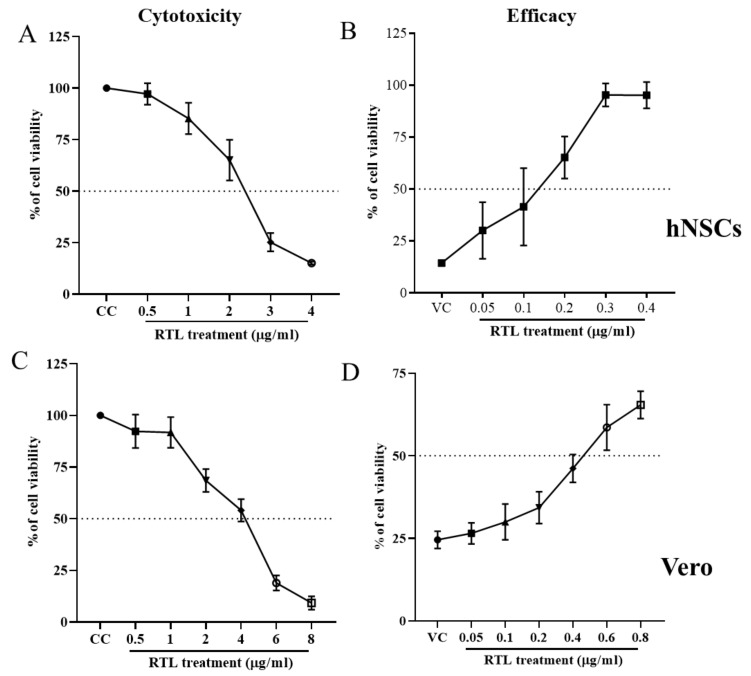
Dose-dependent (**A**,**C**) cytotoxicity and (**B**,**D**) anti-viral efficacy of RTL. (**A**,**B**) hNSCs (2 × 10^4^ cells/well) and (**C**,**D**) Vero cells (1 × 10^4^ cells/well) were plated in 96-well plates and treated (**A**,**C**) with different concentrations of RTL only or (**B**,**D**) infected with ZIKV (MOI 1) then treated with RTL at different concentrations. DMSO (0.1%) was used as vehicle control at 4 dpi, an MTT assay was carried out and the percentage of cell viability was measured by the formula: (sample absorbance/DMSO absorbance) ×100%. Each data point represents the mean ± SD of six wells combined from two independent experiments.

**Figure 4 viruses-13-02317-f004:**
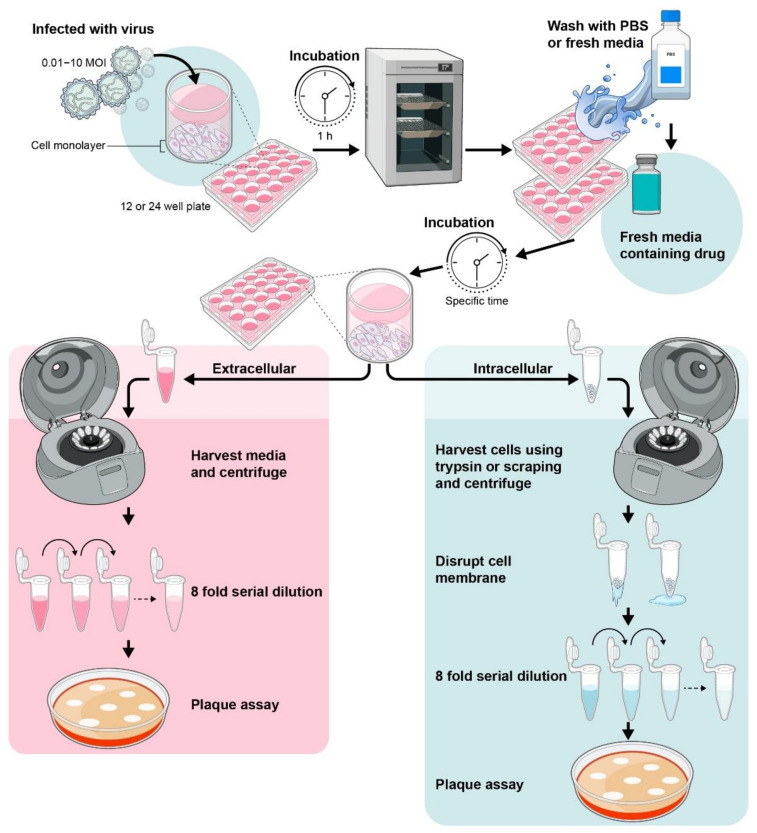
Schematic diagram of measuring antiviral efficacy using plaque assays.

**Figure 5 viruses-13-02317-f005:**
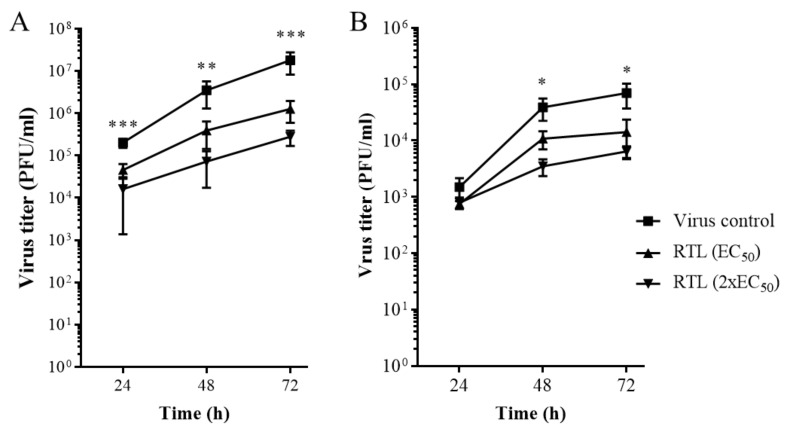
RTL inhibits ZIKV virus replication in hNSCs. (**A**) Extra-cellular and (**B**) intra-cellular virus titer in hNSCs following ZIKV infection and RTL treatment at EC50 and 2xEC50 doses were measured at different time intervals. Each line graph represents the mean ± SD of (**A**) two individual experiments of six individual wells and (**B**) three individual wells of a single experiment. * *p* < 0.05; ** *p* < 0.01; and *** *p* < 0.001 by one-way ANOVA using Dunnett’s multiple comparison test at each time point.

**Figure 6 viruses-13-02317-f006:**
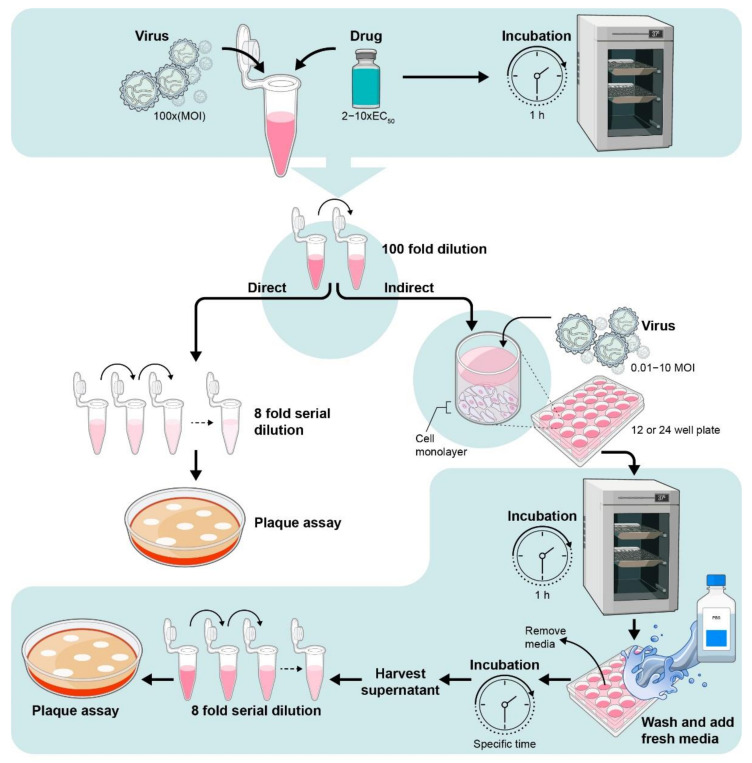
Schematic diagram of the virus inactivation assay.

**Figure 7 viruses-13-02317-f007:**
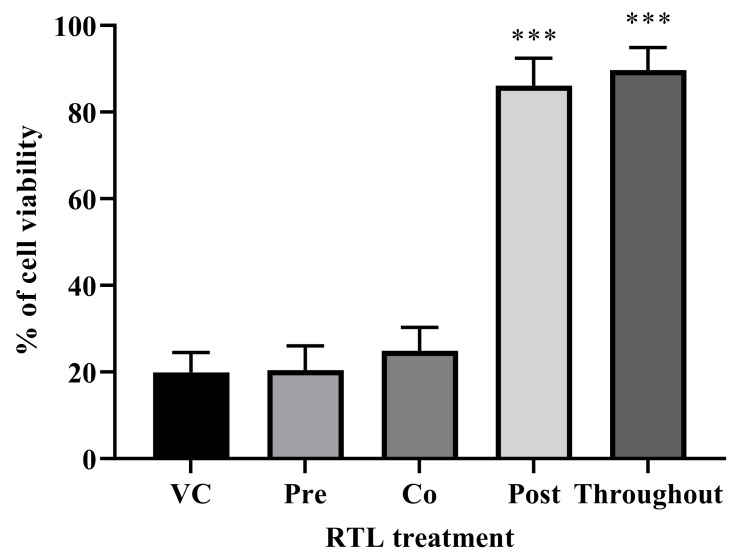
Effect of time of addition of RTL (2xEC50 dose) on ZIKV-induced death of hNSCs. Cells were treated with RTL (2xEC50 dose) at pre-, co-, post- and throughout-treatment time frame of ZIKV infection. Each bar represents differences between six individual wells of a two-independent experiment. *** *p* < 0.001 by one-way ANOVA using Dunnett’s multiple comparison test.

**Table 1 viruses-13-02317-t001:** Basic interpretations from time of addition assay.

Time Point(s)	Possible Outcome
Pre-treatment	Inhibit virus attachment by neutralizing host cell receptor. Inhibit by modulating host cell (immune response). Drug may be useful for prophylactic treatment.
Pre- and Co- treatment	Inhibit virus attachment by neutralizing host cell receptor.
Co-treatment	Inhibit virus attachment by neutralizing host cell receptor. Inhibit virus adsorption or immediately after viral entry. Directly inactivate the virus. Potentially useful for prophylactic or therapeutic treatment.
Co- and Post- treatment	Inhibit virus adsorption or immediately after viral entry. Inhibit cell to cell spread.
Post-treatment	Inhibit virus replication. Inhibit virus trafficking. May be useful for therapeutic treatment.
Throughout- treatment	Positive control for all other time of addition as it combines all time points.

## Data Availability

Not applicable.

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
