# Peer review of "Drug-Screening Strategies for Inhibition of Virus-Induced Neuronal Cell Death"

_viruses, 2021, doi:10.3390/v13112317_

Round 1

Reviewer 1 Report

The authors fixed the minor an major issues raised by the reviewers. The manuscript is now ready for publication. Congratulations.

Reviewer 2 Report

Authors have adequately addressed all suggestions and significantly improved the manuscript. I have no further comments. 

Reviewer 3 Report

In the resubmitted manuscript by Ojha and coworkers, the authors have addressed my concerns and improved the manuscript. The protocols, described in the manuscript, are properly described, include relevant literature and are accompanied with good-looking illustrations. In my opinion, this review will be of interest to researchers in the field and, in my opinion, should be accepted in the present form.

This manuscript is a resubmission of an earlier submission. The following is a list of the peer review reports and author responses from that submission.

Round 1

Reviewer 1 Report

Several viruses can cross the blood-brain-barrier and infect various types of cells in the central nervous system. In the present manuscript, the authors describe protocols to test therapeutic compounds to inhibit virus-induced neuronal death and virus production of neurons, neuroblastoma and neuronal stem cell lines. The protocols are detailed and accompanied with nice illustrations. In my opinion, this review will be of interest to broad neuroscience community and should be accepted after the following concerns are addressed:

  1. The authors should cite relevant literature throughout the protocols. For instance, in the introduction, only a handful of papers are cited. The second sentence alone should include at least four or five citations (rather than review papers, primary research papers should be cited) and the third sentence also deserves a citation. I would also suggest citing a paper or two during the protocol descriptions, so that the reader may check how the relevant part of the protocol was used in the real research. Conclusions should also relate to other studies and hence include citations.
  2. The authors should state the minimum number of repeats/isolations for each experiment.
  3. Lines 333-4; “Culture cells in 48 or 96 well plate using 103 to 105 cells per well (depending on well size) and incubate for 4-6 h in 37ºC CO2 incubator.” Please elaborate, how to decide on the appropriate number of cells per well (should they be confluent, separate but dense, not too dense…?).
  4. Explain what the acronym MOI stands for.
  5. Lines 439-40, 448-9; “Remove the media and add fresh media containing 2-to-10-fold EC50 concentration of the test drug.” If I am not mistaken, the word dilution is missing?

Author Response

  1. The authors should cite relevant literature throughout the protocols. For instance, in the introduction, only a handful of papers are cited. The second sentence alone should include at least four or five citations (rather than review papers, primary research papers should be cited) and the third sentence also deserves a citation. I would also suggest citing a paper or two during the protocol descriptions, so that the reader may check how the relevant part of the protocol was used in the real research. Conclusions should also relate to other studies and hence include citations.

We agree with the reviewer and have added the citations to the manuscript.  Our apologies for the lack of citations in the previous version. We have also included relevant research (lines 363-369, 444-448)

2. The authors should state the minimum number of repeats/isolations for each experiment.

We thank the reviewer for this point. We have included a few lines under section 4 that suggests both duplicate/triplicate wells within an assay as well as repeating each assay in triplicate to obtain consistent and reproducible results (lines 351-355)

3. Lines 333-4; “Culture cells in 48 or 96 well plate using 103 to 105 cells per well (depending on well size) and incubate for 4-6 h in 37ºC CO2 incubator.” Please elaborate, how to decide on the appropriate number of cells per well (should they be confluent, separate but dense, not too dense…?).

We have included suggested confluency for each of the assays.

4. Explain what the acronym MOI stands for.

Completed

5. Lines 439-40, 448-9; “Remove the media and add fresh media containing 2-to-10-fold EC50 concentration of the test drug.” If I am not mistaken, the word dilution is missing?

Our apologies, the word higher was missing in that statement. It has been corrected.

Reviewer 2 Report

In the present paper the authors presented the methodology to test compounds for their ability to examine the effectiveness of anti-viral compounds to prevent neuronal cell death. They describe several protocols, including the isolation and culturing of primary neurons, the culturing of neuroblastoma and neuronal stem cell lines, infection of these cells with viruses, treatment of these cells with selected drugs, measuring virus-induced cell death using MTT or XTT reagents, analysis of virus production from these cells, as well as the basic understanding in mode of action.

The paper will be of great interest to readers, it clearly presents all protocols announced in the abstract. Only minor corrections are needed. Please include additional references to improve the introduction section:

Page 1, lines 26-35: Although it is well known that all these viruses trigger encephalitis, it is important to cite appropriate references (of course, you can site reviews).

Page 1, lines 34,34: Many of these viruses directly infect neurons in the CNS and can induce neuronal damage either by causing neuronal death or neuronal dysfunction.«  - include reference (s).

Author Response

We agree with the reviewer and have added the citations to the manuscript as needed.  Our apologies for the lack of citations in the previous version.  

Reviewer 3 Report

The authors present a  rigorous and well-described protocol for the identification of compounds that serve as therapy for encephalitis. However, the lack of results, at least with some test or control compounds with known activity, makes it an unproven procedure, so it is not shown to be useful for these purposes.

Minor issues: Some operations are described in too much detail (materials and methods), even for the description of a protocol (are obvious).

Author Response

The authors present a rigorous and well-described protocol for the identification of compounds that serve as therapy for encephalitis. However, the lack of results, at least with some test or control compounds with known activity, makes it an unproven procedure, so it is not shown to be useful for these purposes.

We appreciate the reviewer’s very valid point. We have added information on different drug studies relevant for these protocols (lines 363-369, 444-448). We have also included appropriate references that use the same or similar protocols, including our recent publication for which these protocols were directly used.

Minor issues: Some operations are described in too much detail (materials and methods), even for the description of a protocol (are obvious).

We have gone through the protocols and have removed parts of the methods that were too detailed.  We did leave in the instruments and reagents as small differences can have a major impact on the growth of these different cells.  There is not a good detailed protocol for doing drug studies in neuronal cells in the literature, so we erred on being more complete

Round 2

Reviewer 3 Report

In my opinion the problem remains the same as in the previous version: the protocol needs to be tested. If it has not been tested, it is not known if it works, and therefore it should not be published yet. If it has been tested in other works already published, it is not an original publication (it has already been published in those works).